# Ocean wave spectra bias correction through energy conservation for climate change impacts

Andrea Lira Loarca[1] and Giovanni Besio[1]

[1]Department of Civil, Chemical and Environmental Engineering, University of Genoa. Via Montallegro 1, 16145, Genoa, Italy

**Correspondence:** Andrea Lira Loarca (andrea.lira.loarca@unige.it)

**Abstract.** A novel bias-adjustment technique for the 2D directional wave spectra is presented, which accounts for the intraannual temporal variability of waves and the conservation of the wave energy integrated parameter and its extreme distribution, allowing for shifts in frequency and direction given by the GCM-RCM climate signal for the complete multimodal energy distribution. This work represents a first attempt to address the biases inherent in GCM-RCMs wave spectra simulations for an assessment of the magnitudes of the projected changes under a climate change scenario. The bias-correction method is applied to a multi-model ensemble of seventeen EURO-CORDEX regional simulations of wave spectra in eleven locations of the Mediterranean Sea. Climate change impacts are assessed by means of the changes between the bias-adjusted ensemble and hindcast wave spectra for mid-century conditions from 2034 until 2060 and end-of-century from 2074 until 2100. Results highlight the need for novel bias-correction techniques that address the complexity of the possible directional and frequency shifts due to climate change, in order to provide an accurate assessment of projected future changes in wave climate.

## 1 Introduction

Projected changes in wave climate due to shifts in atmospheric circulation have significant implications for coastal planning, adaptation, and mitigation strategies. Accurate representation of current and future wave climate, including its multimodal characteristics, is crucial for understanding coastal hazards and designing effective measures (IPCC, 2019; Oppenheimer et al., 2019). Several studies (e.g. Echevarria et al., 2019; Mortlock and Goodwin, 2015; Portilla-Yandún et al., 2016; Villas Bôas et al., 2017; Shimura and Mori, 2019) have highlighted the importance of resolving directional wave spectra to capture the complexity of ocean waves and identify different wave systems lacking in the traditional studies of wave climate variability which focus on integrated parameters such as significant wave height, mean wave period, and mean wave direction. Recently, Lobeto et al. (2021a); Lira-Loarca and Besio (2022) presented studies on projections of 2D direction wave spectra in different locations around the world and in the Mediterranean Sea, respectively, and highlighted the importance of considering the multimodal behavior of waves to better understand future changes in waves due to climate change.

Wave projections under climate change scenarios are usually generated by wave generation and propagation models driven by surface winds from Global Climate Models (GCMs) or high-resolution dynamically downscaled surface winds from Regional Climate Models (RCMs) (Morim et al., 2018; Jacob et al., 2020; Lira-Loarca et al., 2021b). However, systematic biases are present in GCM atmospheric simulations and the RCM downscaling due to factors such as spatial resolution, simplified

physics and parameterizations, internal variability and downscaling processes (Christensen et al., 2008; Teutschbein and Seibert, 2012). Among the different outputs from GCMs and their regional dowscaling GCM-RCMs, wind fields are known for depicting large uncertainties linked to their local-scale effect capture only with very high horizontal distributions, their spatial heterogeneity and the challenge of the climate models to accurately characterize their turbulent nature (Outten and Sobolowski, 2021). When using GCM-RCMs wind field data to force wave climate projections, these biases and uncertainties are inherited, requiring the application of bias adjustment methods to ensure accurate coastal impact projections (Lemos et al., 2020b). Furthermore, wave climate presents varying timescales ranging from decadal, intraannual, and seasonal to storm events, swells, and wind waves exhibiting high temporal variability which should be considered when applying bias correction techniques (Lira Loarca et al., 2023).

While bias correction techniques are widely used in studies involving climatic and hydrological variables such as precipitation and temperature (scalar variables varying in space and time), their application to wave climate is found in a limited to number of studies (Lemos et al., 2020b, a; Costoya et al., 2020; Lobeto et al., 2021b; Lira-Loarca et al., 2021a) and remains a challenging task due to the multivariate behavior and diverse temporal and spatial variability of waves. Wave climate is often defined by the main integrated wave parameters, $H_s$ (significant wave height), $T_m/T_p$ (mean/peak wave period) and $\theta_m/\theta_p$ (mean/peak wave direction), which are intrinsically correlated both in space and time, and present, by themselves, a high temporal variability on different timescales. Lemos et al. (2020b) applied traditional bias-adjustment techniques, such as the delta method, Empirical Quantile Mapping (EQM), and, Empirical Gumbel Quantile Mapping (EGQM) to the univariate scalar variables, $H_s$, and highlighted the performance of EGQM above the others in the characterization of extreme significant wave heights. Regarding the need to account for the temporal variability of wave climate, (Lira-Loarca et al., 2021a) compared the EQM method considering different time periods (full, seasonal, monthly, day-of-year) and highlighted that the need to consider the temporal variability of waves to accurately adjust biased with the EQM-month method providing a good performance in capturing the correlation and interannual temporal variability. When considering 2D directional wave spectra, Lira-Loarca and Besio (2022) applied a simple "delta method" where the seasonal mean of each bin (frequency and direction) was adjusted to match that of the hindcast and discussed that this method fails to account for the 2D energy distribution and possible bin-variability and does not allow the reconstruction of the bias-adjusted time series of directional spectra, pointing to the need of further research that addresses the correction of systematic errors in 2D wave spectra accounting for changes in the different wave systems, their extreme characteristics and temporal variability.

Among the bias adjustment methods, the Distribution Mapping (DM) method, also known as Empirical Quantile Mapping, Probability Mapping, or Quantile-Quantile mapping, and the Empirical Gumbel Quantile Mapping (EGQM) are widely utilized in atmospheric variables due to its flexibility and ability to address extreme values in the distribution (Teutschbein and Seibert, 2012; Déqué, 2007). The aim of this study is to present the SEGDM-month method, a novel bias correction method for the 2D directional wave spectrum that preserves the behavior of the integrated wave energy. The proposed method is based on the conversion of wave spectrum to energy, the correction of wave energy using the Empirical Gumbel Distribution Mapping (EGDM) method applied on a monthly-basis to account for wave climate temporal variability and the reconstruction to 2D

directional wave spectra maintaining the original energy distribution within frequencies and directions. This method has been implemented for 11 locations in the Mediterranean Sea (Fig. 1) where validated hindcast and multimodel GCM-RCM wave spectrum series are available.

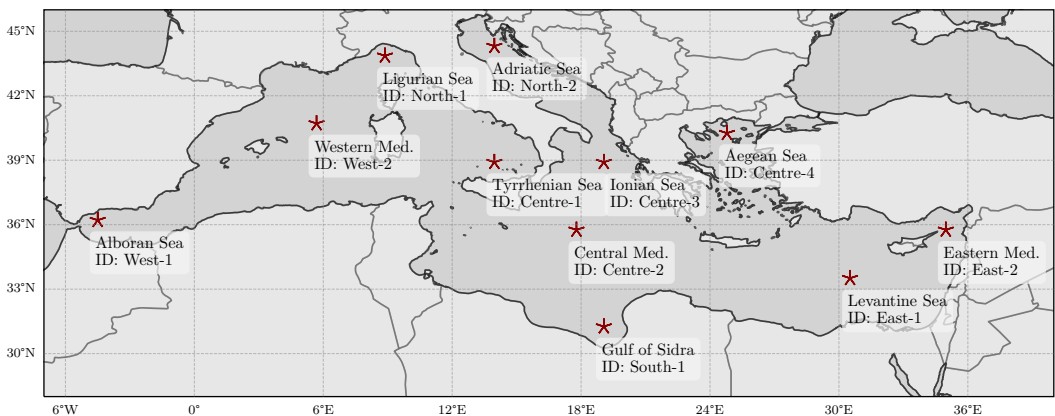

**Figure 1.** Location and identification of the analyzed locations

The manuscript is organized as follows: section "Methods" presents the hindcast and GCM-RCMs wave spectrum datasets used in this study, the skill statistics used to quantify the performance of the GCM-RCMs, and the proposed bias correction
methodology. The results are presented in the section "Results" and discussed in the section "Discussion".

## 2 Methods and Data

### 2.1 Projections of directional wave spectra

The wave spectra hindcast used in this work as proxy for observations was developed by the Meteocean research group[1] of the University of Genoa (Italy), offering hourly high-resolution wave data from 1979 to 2020 on a regular grid with a resolution of
0.127 per 0.09 degrees in longitude and latitude, respectively, equivalent to $\approx 10$ km and hourly 2D directional wave spectrum, $S(f,\theta)$, for eleven locations in the Mediterranean Sea, as shown in Figure 1 and Table 1 (Lira-Loarca and Besio, 2022). These locations cover a variety of sub-basins within the Mediterranean Sea with distinct regional and local wave dynamics (Lazzari et al., 2012; Besio et al., 2016; Di Biagio et al., 2020; Barbariol et al., 2021). The energy spectra of each location is divided into 24 directional bins, $\theta$, of 15 degrees each, and 25 frequency bins, $f$, ranging from approximately 0.07 to 0.66 Hz (or 1.5 to 15
seconds in period). The hindcast has been developed with the third-generation wave model Wavewatch III (version 5.16) (The WAVEWATCH III ® Development Group, 2019) with the growth/dissipation ST4 source terms (Ardhuin et al., 2010; Rascle and Ardhuin, 2013). The hindcast data has been validated using buoy observations in various locations in the Mediterranean Sea. For more information on the setup and validation, refer to (Mentaschi et al., 2013a, b, 2015; Besio et al., 2016).

---

[1]https://meteocean.science

**Table 1.** Analyzed locations for the different representative regions of the Mediterranean sea.

| Point ID | Longitude | Latitude | Region | Approx. Depth [m] |
|----------|-----------|----------|--------|-------------------|
| West-1 | -4.5 | 36.21 | Alboran Sea | 1000 |
| West-2 | 5.69 | 40.71 | Western Mediterranean | 2700 |
| North-1 | 8.87 | 43.86 | Ligurian Sea | 1240 |
| Centre-1 | 13.96 | 38.91 | Tyrrhenian Sea | 3400 |
| North-2 | 13.96 | 44.31 | Adriatic Sea | 60 |
| Centre-2 | 17.78 | 35.76 | Central Mediterranean | 4000 |
| Centre-3 | 19.06 | 38.91 | Ionian Sea | 1360 |
| South-1 | 19.06 | 31.26 | Gulf of Sidra | 830 |
| Centre-4 | 24.79 | 40.26 | Aegean Sea | 780 |
| East-1 | 30.52 | 33.51 | Levantine Sea | 2600 |
| East-2 | 34.97 | 35.76 | Eastern Mediterranean | 1100 |

Wave climate projections were obtained with the same WW3 configuration forced by surface wind fields of seventeen Euro-CORDEX (Table 2) (Jacob et al., 2014, 2020) models (GCM-RCM combinations) with a temporal resolution of 6 hours (Lira-Loarca et al., 2021b; De Leo et al., 2021; Lira-Loarca and Besio, 2022). Wave integrated parameters and 2D directional spectra for the 11 locations were obtained for each GCM-RCMs for the base-period (1970-2005) and RCP8.5 scenario (2006 - 2100). For details on the definition and performance of the different RCMs used in this work, the reader is referred to Strandberg et al. (2014) for the Rossby Centre regional climate model RCA4, Will et al. (2017) for the CLM-Community CCLM4-8-17 model, Christensen et al. (2007) for the Danish Climate Centre regional climate model HIRHAM5 and Leutwyler et al. (2017) for the COSMO-CLM accelerated version COSMO-crCLIM-v1-1.

The integrated parameter, wave energy $E$ [m$^2$s/deg], is defined as,

$$E = \int\limits_{0}^{2\pi} \int\limits_{0}^{\infty} S(f,\theta) \, df \, d\theta, \tag{1}$$

where $S(f,\theta)$ is the directional wave spectrum.

The contribution of a spectral bin $(f, \theta)$ to the directional wave spectrum is defined as,

$$\widehat{S(f,\theta)} = \frac{S(f,\theta)\Delta f \Delta \theta}{E}, \tag{2}$$

### 2.2 Skill statistics

Skill statistics ($Bias$ and $RMSE$) have been computed for the monthly means and maxima of the directional wave spectral density function between the GCM-RCM data and the hindcast for the common period of $1979 - 2005$ for all the locations and models presented in Tables

**Table 2.** Notation of the used EURO-CORDEX GCM-RCM datasets

| GCM | RCM | | | |
|---|---|---|---|---|
| | CCLM4-8-17 | RCA4 | HIRHAM5 | COSMO-crCLIM-v1-1 |
| CCCma-CanESM2 | CCLM4-CanESM2 | | | |
| MIROC-MIROC5 | CCLM4-MIROC5 | | | |
| MPI-M-MPI-ESM-LR | | RCA4-MPI-ESM-LR | HIRHAM5-MPI-ESM-LR | |
| NCC-NorESM1-M | | RCA4-NorESM1-M | HIRHAM5-NorESM1-M | COSMO-crCLIM1-NorESM1-M |
| CNRM-CERFACS-CNRM-CM5 | | RCA4-CNRM-CM5 | HIRHAM5-CNRM-CM5 | |
| IPSL-IPSL-CM5A-MR | | RCA4-IPSL-CM5A-MR | HIRHAM5-IPSL-CM5A-MR | |
| MOHC-HadGEM2-ES | | RCA4-HadGEM2-ES | HIRHAM5-HadGEM2-ES | COSMO-crCLIM1-HadGEM2-ES |
| ICHEC-EC-EARTH | | RCA4-EC-EARTH | HIRHAM5-EC-EARTH | COSMO-crCLIM1-EC-EARTH |

$$Bias = \frac{1}{N} \sum_{i=1}^{N} \left( \overline{S(f,\theta)}_{RCM_i}^{baseline} - \overline{S(f,\theta)}_{hind_i} \right), \tag{3}$$

$$RMSE = \sqrt{\frac{1}{N} \sum_{i=1}^{N} \left( \overline{S(f,\theta)}_{RCM_i}^{baseline} - \overline{S(f,\theta)}_{hind_i} \right)^2}, \tag{4}$$

where $\overline{S(f,\theta)}$ is the seasonal mean directional wave spectrum, $N$ is the length of the dataset and the sub/superscripts $hind$ and
$RCM|baseline$ correspond to the hindcast and GCM-RCM baseline simulations, respectively, from 1979 to 2005.

## 2.3 Bias correction

The bias-adjustment methodology consists on i) the conversion of 2D directional wave spectra to wave energy, ii) the correction
of the wave energy 3-hour time series using the EGDM-month method and iii) the reconstruction of the wave spectrum con-
serving the initial energy distribution in the frequency and directional bins. The bias correction method proposed in this study,
Spectral Energy Gumbel Distribution Mapping per month (SEGDM-month), is based on the well-known Distribution Mapping
(also known as Empirical Quantile Mapping) method (Déqué, 2007) and its adaptation for the correction of the upper-tail of
the distribution using a Gumbel parametric distribution (Lemos et al., 2020b). To account for the temporal variability of the
wave climate, correction is done on a month-by-month basis, correcting each month independently (Lira Loarca et al., 2023).
The DM method consists in the adjustment of the distribution of the GCM-RCM projections to match the distribution of the
reference historical data,

$$E^* = F_{hind}^{-1} \left( F_{RCM}^{baseline}(E) \right), \tag{5}$$

where $E^*$ is the bias-adjusted energy, $F_{ref}$ is the distribution function of the reference historical data (hindcast, $1979 - 2005$)
and $F_{RCM}^{baseline}$ is the distribution function of each GCM-RCM during the baseline period ($1979 - 2005$). The correction is

implemented using the Empirical Cumulative Distribution of the datasets for the lower tail and mean body of the distribution
(up to the 90*th* quantile) with linearly distributed quantiles $[q_1, q_{90}]$ every 1. The upper tail of the distribution (above the 90*th*
quantile) is corrected according to the bias between the parametric Gumbel distributions of the GCM-RCM and reference data.

Once the energy of the integrated parameter, $E$, has been bias corrected, the conversion to the directional wave spectrum is
done, conserving the relative contribution of each spectral bin $(f, \theta)$ in the original wave spectrum $(\widehat{S(f,\theta)}$, Eq. 2).

$$S^*(f,\theta) = E^* \cdot \widehat{S(f,\theta)}, \tag{6}$$

where $S^*(f,\theta)$ is the bias-corrected directional wave spectrum.

## 2.4 Performance of bias correction methods

To measure the effectiveness of the bias correction method, simple delta methods have been employed for both the $RMSE$
and $Bias$, defined as,

$$\Delta RMSE = RMSE^* - RMSE^{raw} \quad \text{and} \quad \Delta Bias = |Bias^*| - |Bias^{raw}|, \tag{7}$$

where the $*$-superscript denotes the bias-corrected metric and the $raw$-superscript denotes the raw metric (without bias correc-
tion). Therefore, negative $\Delta$ indicates better performance of the bias-adjusted data with respect to the raw data.

## 2.5 Projected changes in the 3D directional wave spectra

The assessment of future changes in the directional wave spectra under RCP8.5 has been done through the analysis of seasonal
differences between the multi-model ensemble mean and the hindcast wave spectra. For the calculation of the multi-model
ensemble mean, a weighted approach is used:

$$WE[S(f,\theta)] = \frac{\sum_{m=1}^{N_m} w_m \cdot \overline{S^*(f,\theta)_m}}{\sum_{m=1}^{N_m} w_m}, \tag{8}$$

where $\overline{S^*(f,\theta)}_m$ refers to the bias-adjusted seasonal mean energy density for each GCM-RCM ($m = 1 \dots 17$), $WE[S(f,\theta)]$
represents the seasonal weighted multi-model ensemble mean, $N_m$ is the number of GCM-RCMs included in the ensemble
($N_m = 17$), and $w_m$ denotes the corresponding weight for each GCM-RCM, which is calculated based on the number of
ensemble members forced with the same GCM in relation to $N_m$. An ordinary arithmetic ensemble mean was also calculated,
yielding similar results to the weighted mean approach.

To assess the projected changes in directional wave spectra between the GCM-RCM simulations for the future RCP8.5 and
baselined scnearios, the relative change is evaluated between the multi-model ensemble seasonal mean for both periods:

$$\Delta S(f,\theta) = WE[S(f,\theta)]_{RCP8.5} - WE[S(f,\theta)]_{baseline}, \tag{9}$$

where $\Delta S(f,\theta)$ represents the seasonal projected relative change and $\overline{S(f,\theta)}_{hind}$ corresponds to the hindcast seasonal mean.

## 3 Results

### 3.1 Performance of bias-adjusted 2D directional wave spectra

The performance of the SEGDM-month bias-adjustment method is first assessed through the correction of the integrated wave
energy parameter. Figure 2 presents, for each panel, the monthly variability of different energy percentiles for the hindcast
(dotted line) and the multi-model ensemble mean (solid line). The ensemble uncertainty is quantified with one standard deviation (shaded region). The rows correspond to different locations, while the columns present the raw (left) and bias-adjusted
(right) GCM-RCM ensemble. The bias in the wave energy monthly distribution of the raw GCM-RCMs is higher for the higher
percentiles both in magnitude and in the ability to represent the temporal variability. The raw GCM-RCMs presented an overestimation of the upper-quantiles for the winter months for North-1 (Ligurian Sea), West-2 (Western Mediterranean) and East-1
(Levantine Sea) and an underestimation for North-2 (Adriatic Sea) and West-1 (Alboran Sea) which are adequately corrected
by the SEGDM-month method for all locations, advocating the use of bias-correction methods for integrated parameters. Additionally, the use of the Gumbel distribution for the upper-tail allows for a correct characterization of the energy extremes of
the bias-corrected GCM-RCMs simulations depicted by the higher percentiles.

In order to understand the performance of the GCM-RCMs against the hindcast during the baseline period $(1979 - 2005)$
for the 2D directional wave spectra, the seasonal $Bias$ (Eq. 3) and $RMSE$ (Eq. 4) of the monthly mean values of the wave
spectrum are computed. We present the results of $Bias$ for all seasons to analyze the capacity of the method in capturing the
intraannual temporal variability of waves. Due to space limitation, we only present the $RMSE$ during winter (DJF), where
the highest wave energy systems are expected. The remaining seasonal $RMSE$ results are presented in the *Supplementary
Information*.

Figure 3 presents for each analyzed location the winter (DJF) mean of the wave spectrum monthly means for the hindcast
(left), the $Bias$ between the hindcast and the raw GCM-RCM (middle) and the difference in bias, $\Delta Bias$ (Eq. 7), for the
bias-adjusted GCM-RCM (right). For each location, the worst-performing GCM-RCM is presented. Figures 4 to 7 present, for
the monthly means, the winter $RMSE$ and the spring, summer and fall $Bias$, respectively.

For winter $Bias$ of the monthly means (Fig. 3) the results show a systematic underestimation for the most energetic bins for
the raw GCM-RCM spectra in the locations West-1 (Alboran Sea), West-2 (Western Mediterranean), North-2 (Adriatic Sea)and
Centre-4 (Aegean Sea), while the remaining locations present an overestimation of the most energetic bins which correspond
to the swell system. Negative $\Delta Bias$ indicates an improvement in the performance of the GCM-RCM wave spectra with
respect to the hindcast due to bias correction. It can be observed that the bias, in the most energetic systems, is reduced for
all locations, except for the North-2 (Adriatic Sea) and Centre-4 (Aegean Sea), where no noticeable changes are observed and
Centre-3 (Ionian Sea), where the most energetic system SE-SW presents decreases for the SW waves and increases for the SE
waves. It can also be observed that for some locations, the use of bias-correction method leads to an increase in bias for the
least energetic systems, as observed for West-1 (Alboran Sea), Centre-3 (Ionian Sea), North-2 (Adriatic Sea), West-2 (Western

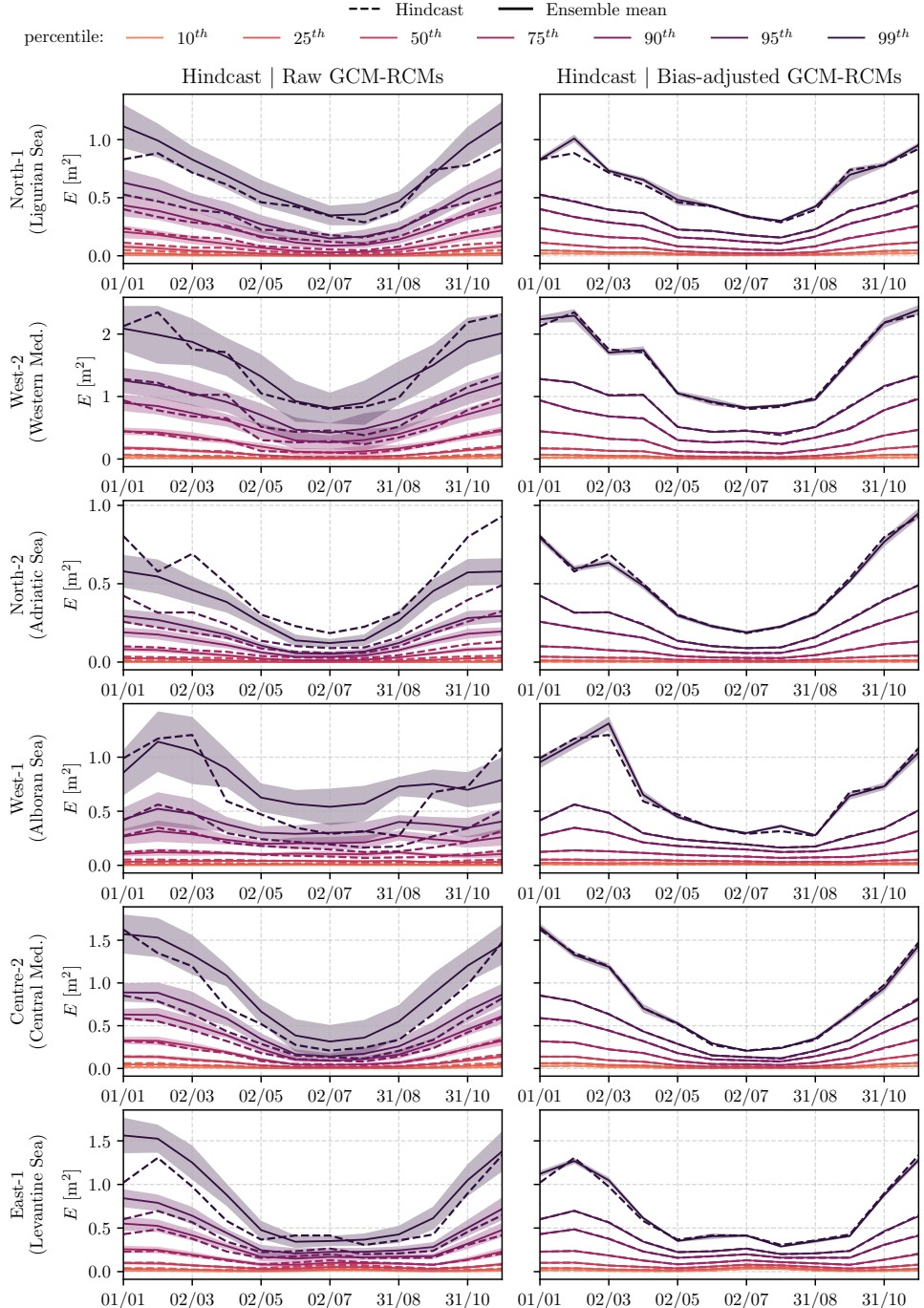

**Figure 2.** Energy $E$ [m$^2$] monthly percentiles (10, 25, 50, 75, 90, 95, 99th) for the hindcast (dotted line) and the multi-model ensemble mean (solid line) and one-standard deviation from the mean (shaded region) for the baseline period (1979 − 2005). The rows present the different analyzed locations: North-1 (Ligurian Sea), West-2 (Western Mediterranean), North-2 (Adriatic Sea), West-1 (Alboran Sea), Centre-2 (Central Mediterranean) and East-1 (Levantine Sea), whereas the columns correspond to the raw and bias-adjusted GCM-RCMs data.

Mediterranean) and Centre-4 (Aegean Sea), where increases in biases ranging from 0.002-0.005 m$^2$s/deg but the integrated wave energy is conserved to the hindcast distribution.

Regarding the winter mean of the $RMSE$ of the monthly means depicted in Figure 4 it can be observed that the use of the bias correction method presented in this work leads to increased error in the North-2 (Adriatic Sea) and Centre-4 (Aegean Sea) locations for the winter RMSE, although different behaviors are observed for the remaining seasons (*Supplementary Information*). Indeed the North-2 (Adriatic Sea) and Centre-4 (Aegean Sea) locations are characterized by local scale phenomena that might not be entirely captured by GCM-RCM resolution (Lira-Loarca and Besio, 2022). In the case of the West-1 (Alboran Sea), Centre-3 (Ionian Sea), and West-2 (Western Mediterranean) where the $Bias$ presented a decrease of performance for the bias-adjusted low energy systems, the $RMSE$ depicts an increase of performance for all bins. This is due to the calculation of the $RMSE$ for wave spectra errors which exhibit values less than one, and upon squaring them, their magnitudes further diminish leading to smaller errors for the $RMSE$ with respect to $Bias$. For the remaining locations, the application of the bias-adjustment techniques leads to an improved performance of the data for all spectral bins.

Regarding the bias-correction results for the remaining seasons, Figures 5 to 7 present the $Bias$ of the monthly means for spring, summer and fall, respectively. A similar behavior is depicted for all seasons, where the bias correction leads to improvement in performance for the most energetic wave systems. Spring (MAM, Fig. 5) and fall (SON, Fig. 7) present similar spectra distribution for hindcast as winter, but contrary to winter, which has been described previously and in which the bias-correction led to an increase in errors for some locations, a general improvement in the performance is observed for both spring and fall, for all locations. For example, the West-2 (Western Mediterranean), West-1 (Alboran Sea), and Centre-4 (Aegean Sea) locations presented an increase in errors in the least energetic systems after the bias-correction during winter, but this is not the case for spring and fall where there is a general improvement of all systems in these locations. Although the hindcast energy density magnitude for spring and fall is smaller than in winter, decreases in $Bias$ of around 50% are observed for all locations and systems. A similar behavior is depicted for the summer Mean $Bias$ (Fig. 6) where the bias correction leads to an improvement in performance in almost all locations with the exception of North-2 (Adriatic Sea) and East-1 (Levantine Sea) for the low energy bins. Therefore, the use of the monthly-EGQM method presented in this work captures the intra-annual temporal variability of the spectra and correctly adjusts the bias for all seasons.

## 3.2 Future changes in 2D wave spectra under RCP8.5

Regarding the future changes under the RCP8.5 climate change scenario, Figures 8 and 9 present the projected changes for future monthly means and maxima, respectively, during spring (MAM). The seasonal monthly maxima is the average of the monthly maximum of the 2D directional wave spectra during the analyzed period where the selection of the monthly maxima spectra is done corresponding to the point in time of the maximum wave energy. The remaining seasons are included in the *Supplementary Information*. Each figure presents, for each location, the spring (MAM) mean of the wave spectrum monthly means/maxima for the hindcast and multi-model ensemble mean during the baseline period $(1979 - 2005)$ and the differences between the multi-model ensemble mean for mid-century $(2034 - 2060)$ and end-of-century $(2074 - 2100)$ with respect to the

baseline period. Stippling indicates regions where at least 80% of the models agree on the sign of the change. Lack of stippling indicates low model agreement (less than 80%).

Regarding the projected changes in the monthly mean wave spectra (Fig. 8) it can be observed that the West-1 (Alboran Sea) presents similar 2D spectra between hindcast and the ensemble mean for baseline conditions for the most energetic Easterly waves with robust projected increases for both periods, leading to a projected change of bi-modal hindcast spectra to predominantly unimodal Easterly swells. For the North-1 (Ligurian Sea) location it can be highlighted that the ensemble presents higher spectra values with respect to hindcast for the main SW system and projections indicate different behaviors in this system, depending on the frequency although without model agreement. The Centre-3 (Ionian Sea) location presents projected decreases for the more energetic Southeasterly in the lower frequencies for both periods and an increase for the higher frequencies.

The West-2 (Western Mediterranean) location depicts a future behavior in which the main hindcast NW-N swell system presents an increase in both periods, although without model agreement and a robust decrease in the less energetic SW waves, which were not noticeable in the hindcast. This behavior is also observed in the North-2 (Adriatic Sea) location where a robust increase in the SE waves is depicted for both periods. On the other hand, a different behavior is observed for the remaining locations where the projected changes indicate a robust decrease in the most energetic swell system. For the Centre-2 (Central Mediterranean) and South-1 (Gulf of Sidra)  locations, an increase can be observed for the less energetic systems, although without model agreement.

Regarding the projected changes in the monthly maxima wave spectra during spring (Fig. 9), a projected increase, without model agreement, is observed in the main SE swell system of the North-1 (Ligurian Sea) leading to more intense southwestern storm events. This behavior is also observed for the West-1 (Alboran Sea), North-2 (Adriatic Sea) and West-2 (Western Mediterranean) locations, where an increase in the main system is observed for future conditions. The Centre-3 (Ionian Sea), provides different results with a main SE-SW hindcast swell system that depicts robust projected decreases for the NW waves and increases for the SW-W waves for the higher frequencies, leading to a more bimodal or unimodal behavior, in contrast to the hindcast multimodal extreme wave spectra. For the Centre-2 (Central Mediterranean) and Centre-4 (Aegean Sea)locations, different behavior are observed for the varying frequencies and directions of the spectra with no definite results on the future behavior of the systems. Finally, the Centre-1 (Tyrrhenian Sea), South-1 (Gulf of Sidra) , East-1 (Levantine Sea) and East-2 (Eastern Mediterranean) present robust decreases in the main energetic systems for both future periods. It can be highlighted that, for all locations, the ensemble mean during baseline conditions provides an accurate representation of the hindcast wave spectra, highlighting the performance of the SEGDM-month method and the energy distribution provided by the GCM-RCMs.

## 4 Discussion and Conclusions

This work presents the application of a novel bias-correction technique designed for 2D wave spectra by means of the conservation of the integrated wave energy. Although the application of bias correction techniques is extended among hydrological impact studies and the need and benefits in wave projections have been highlighted by recent studies applied to integrated wave parameters (Lemos et al., 2020b, a; Lira Loarca et al., 2023), the correction of the 2D directional wave spectra considering the frequency and directional dimensions integrally remains unexplored (Lobeto et al., 2021a; Lira-Loarca and Besio, 2022). Lira-Loarca and Besio (2022) performed bias adjustment by applying the widespread delta method to the seasonal mean statistics of the wave spectra on a bin-by-bin basis, which did not allow for a reconstruction of the complete time series of spectra, could lead to a lack of energy conservation within the bias-corrected spectra, and, is not flexible in allowing possible future directional and frequency shifts that were not present in the historical baseline conditions. This study is a first attempt to perform bias-adjustment of 2D directional wave spectra not bin-by-bin but in an integrated frequency-direction manner. The SEGDM-month works on the correction of the integrated wave energy parameter taking into account the intra-annual variability and allows the energy distribution in frequency and direction following the climate signal of the GCM-RCM. The choice to maintain the energy distribution given by the original GCM-RCM allows one to explore the possible shifts in direction and periods given by the global climate models, not present in historical conditions. In order to compare the possible implications of the bias correction technique on the assessment of future changes, Figure 10 presents the mean directional spectra for spring (March, April and May) obtained by Lira-Loarca and Besio (2022) and the ones presented in this manuscript. It can be observed that for the West-1 (Alboran Sea) and Centre-3 (Ionian Sea) locations, both the delta and SEGDM-month method lead to a similar distribution for changes under mid-century conditions, whereas some changes present in the delta-method for end-of-century, such as robust decrease for the western system in the West-1 (Alboran Sea), and, increases in the eastern directions for Centre-3 (Ionian Sea), are not present for the SEGDM-month method. On the other hand, for the North-1 (Ligurian Sea), different, almost contrary, results are obtained within the two methods, where an increase north swells was depicted for end-of-century for the delta-method, not present in the SEGDM-month, where both decreases and increases are obtained for the main SW system. Therefore the choice of a bias-correction method is crucial for a correct assessment of future changes and consequently, coastal and marine adaptation and resilient strategies. This work highlights the need to consider possible future shifts in the 2D wave spectra for local impact assessments instead of only considering changes in integrated parameters such as significant wave height and mean wave direction.

The SEGDM-month bias correction method presented in this work allows the correction of the full time series of the 2D directional wave spectra allowing for a better characterization of extremes due to the fit of a Gumbel distribution to the upper tail of the energy distribution, which, to the authors' knowledge, has not been addressed in previous studies. Figure 11 presents for each location analyzed the winter (DJF) mean of the wave spectrum monthly maxima for the hindcast (left), the $RMSE$ between the hindcast and the raw GCM-RCM (middle) and the difference in bias, $\Delta RMSE$ (Eq. 7), for the bias-adjusted GCM-RCM (right). For each location, the worst-performing GCM-RCM is presented. It can be seen that for all locations, except North-2 (Adriatic Sea) and Centre-4 (Aegean Sea), there is an improvement in the performance of the baseline simula-

tion of GCM-RCM highlighting the ability of the proposed method to adjust the biases for different 2D spectra distributions. Following these results, further studies could focus on multivariate bias-corrections techniques although care must be taken to allow possible bin-shifts given by future climate signal and to account for the wave temporal variability and extreme events.

This study focuses on bias correction and future changes in directional wave spectra, relying exclusively on the high-emission, business-as-usual scenario RCP8.5. This approach provides insights into future changes and risks under a plausible yet pessimistic scenario. However, the use of a single scenario is recognized as a limitation for developing mitigation strategies, as incorporating multiple emission and mitigation pathways would enable a more comprehensive assessment of projected wave climate changes. The decision to use only RCP8.5 was driven by constraints in computational resources and data storage, which are necessary for simulating a large ensemble of GCM-RCM wave climate projections. Additionally, within EURO-CORDEX, the availability of GCM-RCM forcings is significantly greater for RCP8.5 than for other scenarios. Future research should address this limitation by incorporating a broader range of emission and mitigation scenarios, ensuring an optimal balance between scenarios, GCMs, and RCMs.

This work presents a step forward in the analysis of 2D wave spectra projections, which allows identifying changes in individual wave systems, allowing for a more detailed and realistic assessment of future wave changes with respect to the use of integrated wave parameters where the use of mean or peak values does not allow for a full understanding of the complexities of wave climate dynamics.

*Author contributions.* G.B conceptualized the study, A.L.L developed the methodology, performed the analysis and prepared the visualization and the original draft of the manuscript, A.L.L and G.B. reviewed and edited the manuscript, and administrated and acquired the financial support for the projects leading to this publication.

*Competing interests.* The authors declare no competing interests.

*Acknowledgements.* A.L.L. acknowledges the FOCUSMed project - Young Researchers Seal of Excellence grant awarded by the Italian Ministry of University and Research.

G.B. acknowledges the RETURN Extended Partnership and received funding from the European Union Next-GenerationEU (National Recovery and Resilience Plan – NRRP, Mission 4, Component 2, Investment 1.3 – D.D. 1243 2/8/2022, PE0000005).

The authors acknowledge the CINECA ISCRA-C IsC87-UNDERSEA, IsC87-FUWAMEBI, IsCa2-COCORITE and IsCa2-EXWAMED projects for the computing resources to perform the GCM-RCMs wave projections and post-processing.

The authors thank the developers of the scientific software that enabled this study, namely `xarray` (Hoyer and Hamman, 2017), `wavespectra` (Guedes et al., 2021) and `xclim` (Logan et al., 2021).

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

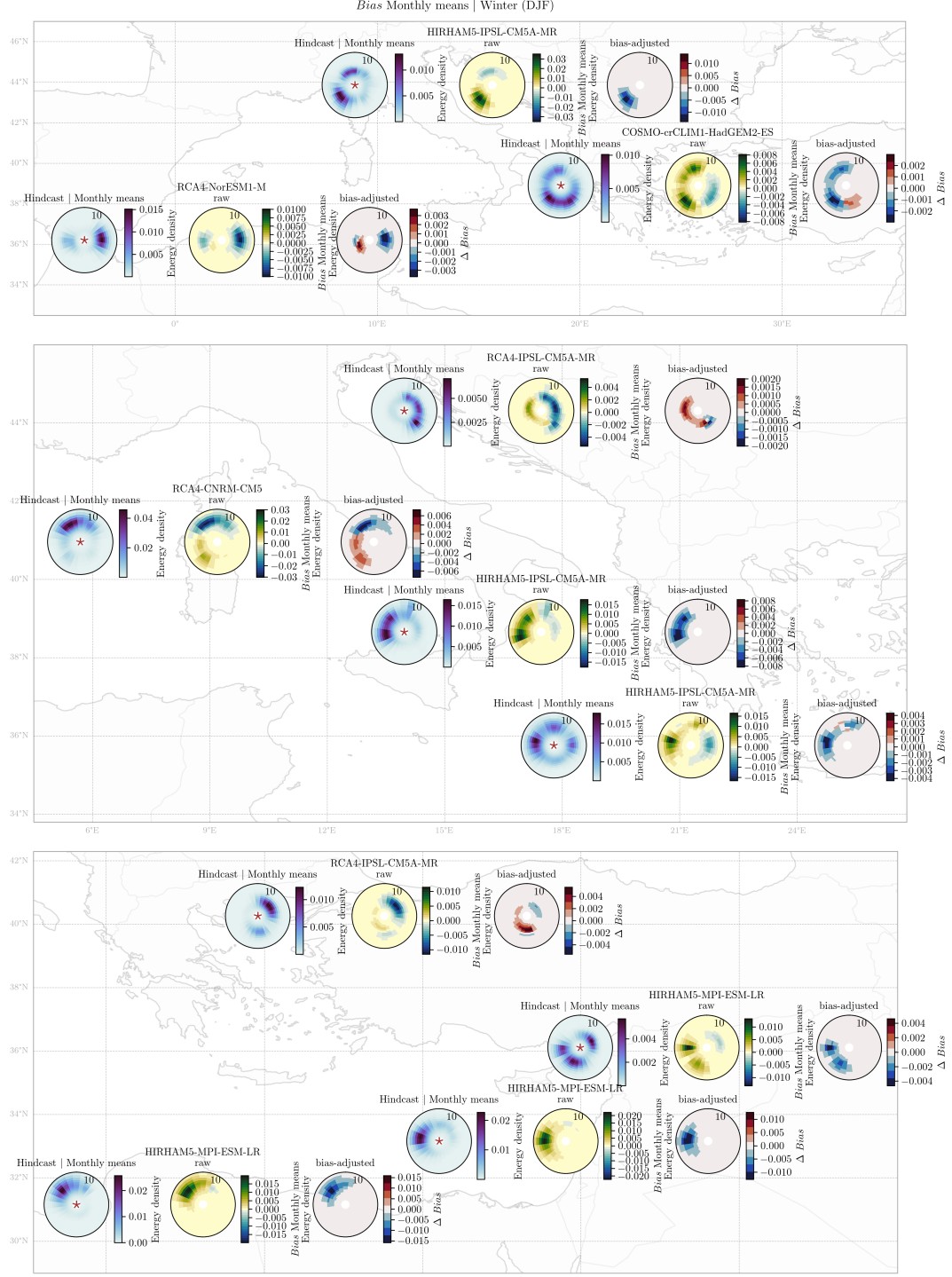

**Figure 3.** Results of the monthly mean 2D spectra averaged over the winter (DJF), in m$^2$s/deg. For each location: hindcast wave spectrum (left panel), $Bias$ between the raw worst-performing GCM-RCM and hindcast (middle) and $\Delta Bias$ between the bias-adjusted $Bias^*$ and raw $Bias$. The star in the hindcast/left panel represents the location of the points corresponding to Figure 1.

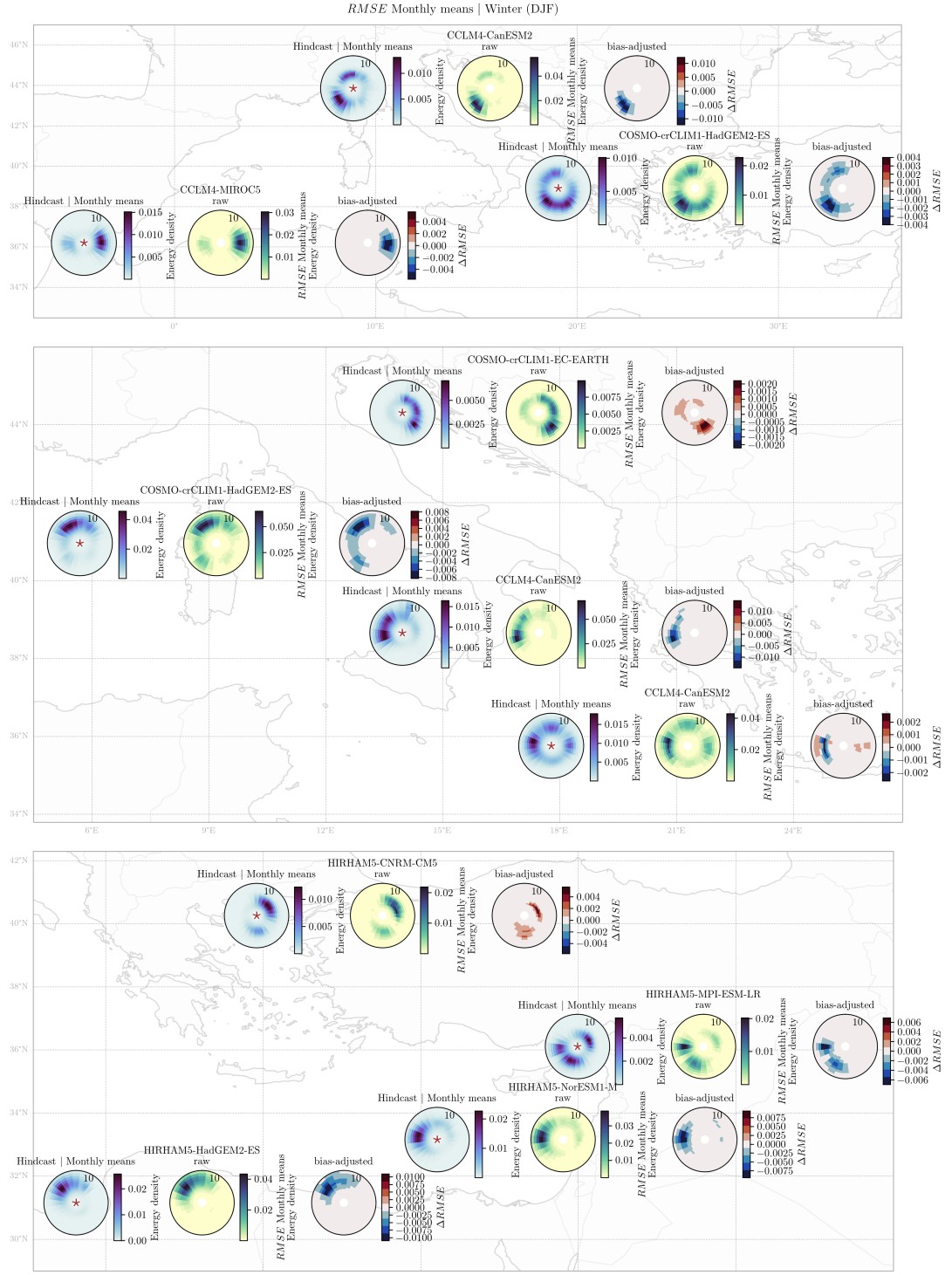

**Figure 4.** Results of the monthly mean 2D spectra averaged over the winter (DJF), in m$^2$s/deg. For each location: hindcast wave spectrum (left panel), $RMSE$ between the raw worst-performing GCM-RCM and hindcast (middle) and $\Delta RMSE$ between the bias-adjusted $RMSE^*$ and raw $RMSE$.

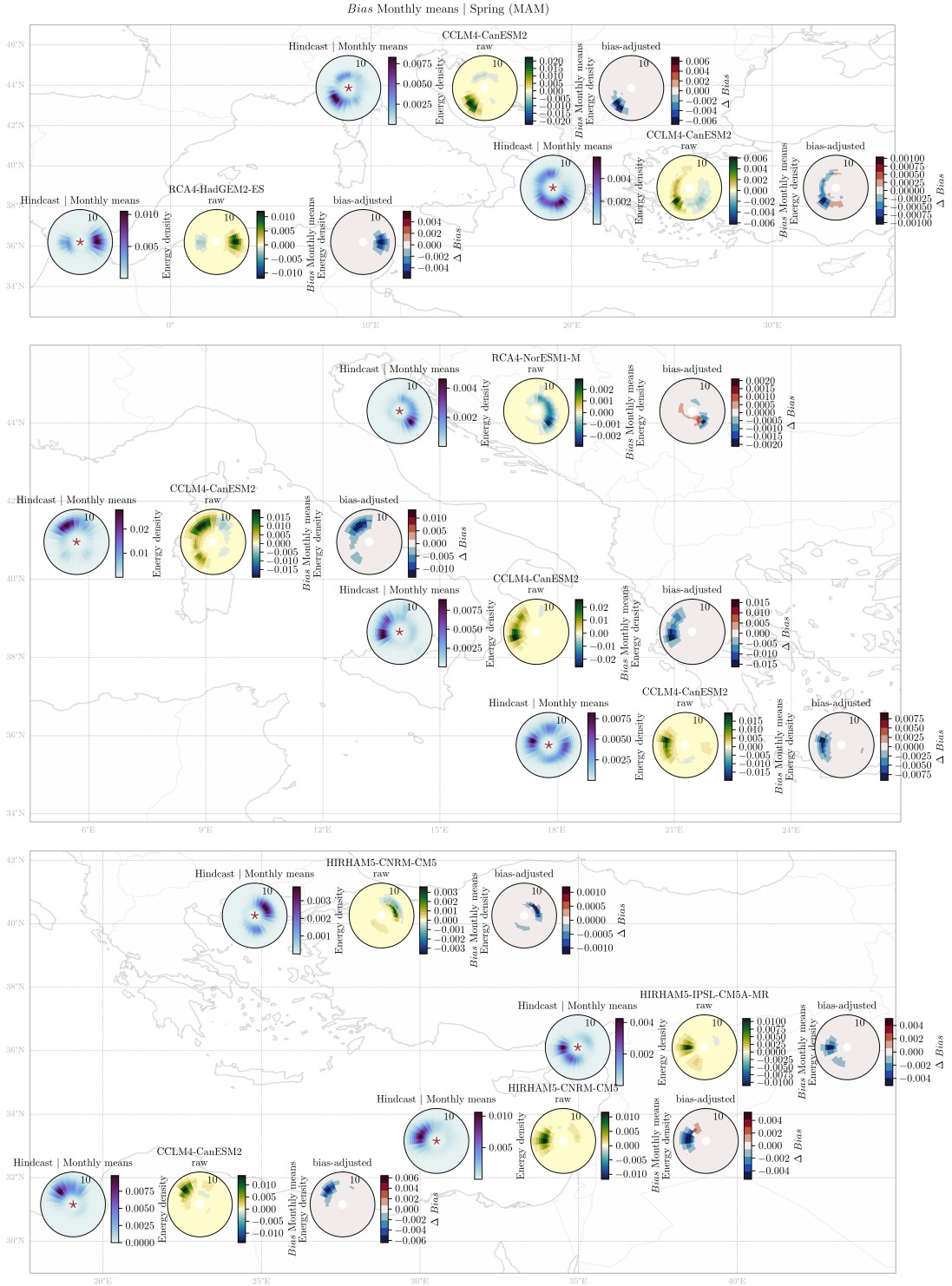

**Figure 5.** Monthly mean 2D spectra averaged over the spring (MAM), in m$^2$s/deg. For each location: hindcast wave spectrum (left panel), $Bias$ between the raw worst-performing GCM-RCM and hindcast (middle) and $\Delta Bias$ between the bias-adjusted $Bias^*$ and raw $Bias$.

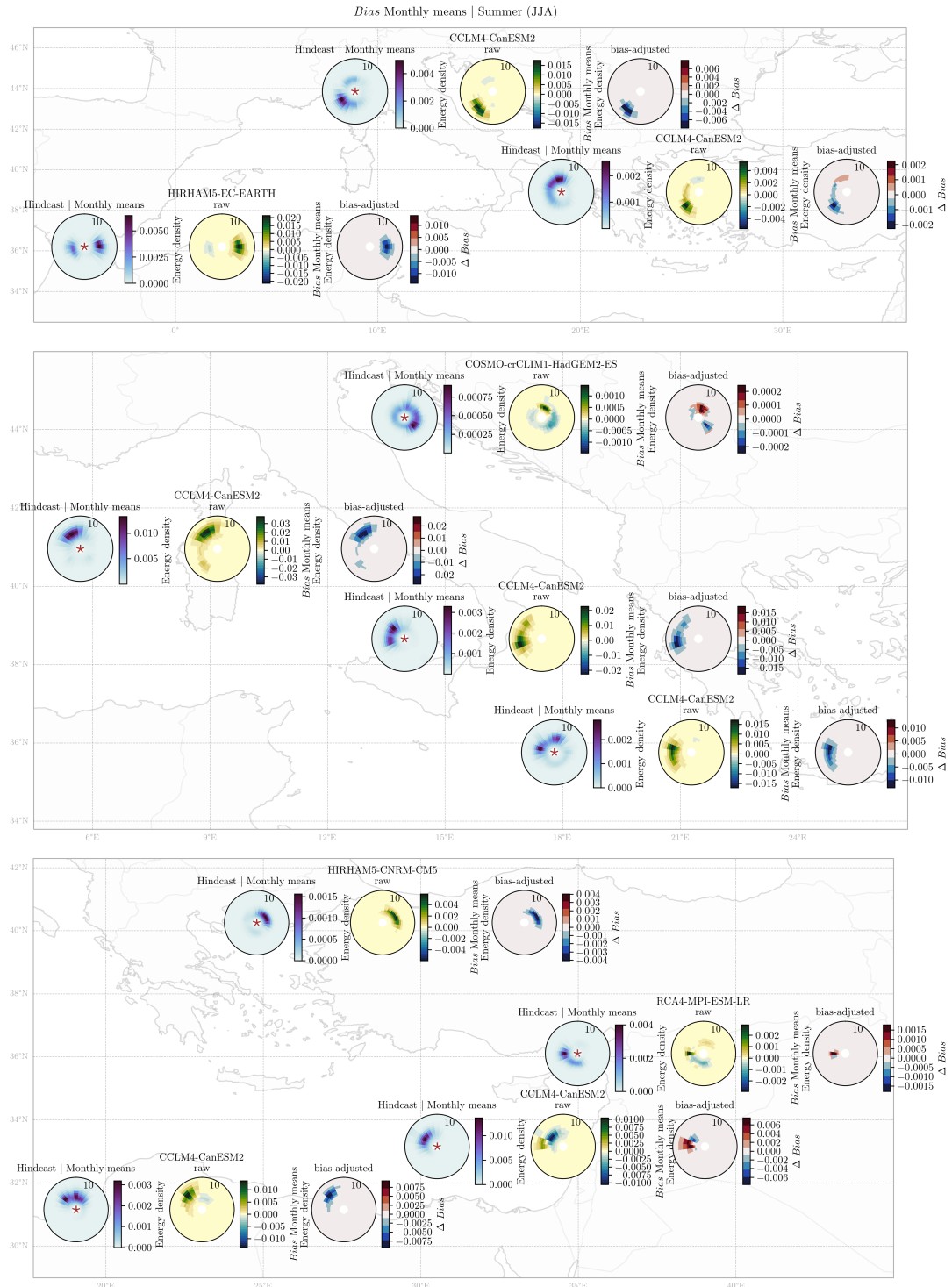

**Figure 6.** Monthly mean 2D spectra averaged over the summer (JJA), in m$^2$s/deg. For each location: hindcast wave spectrum (left panel), $Bias$ between the raw worst-performing GCM-RCM and hindcast (middle) and $\Delta Bias$ between the bias-adjusted $Bias^*$ and raw $Bias$.

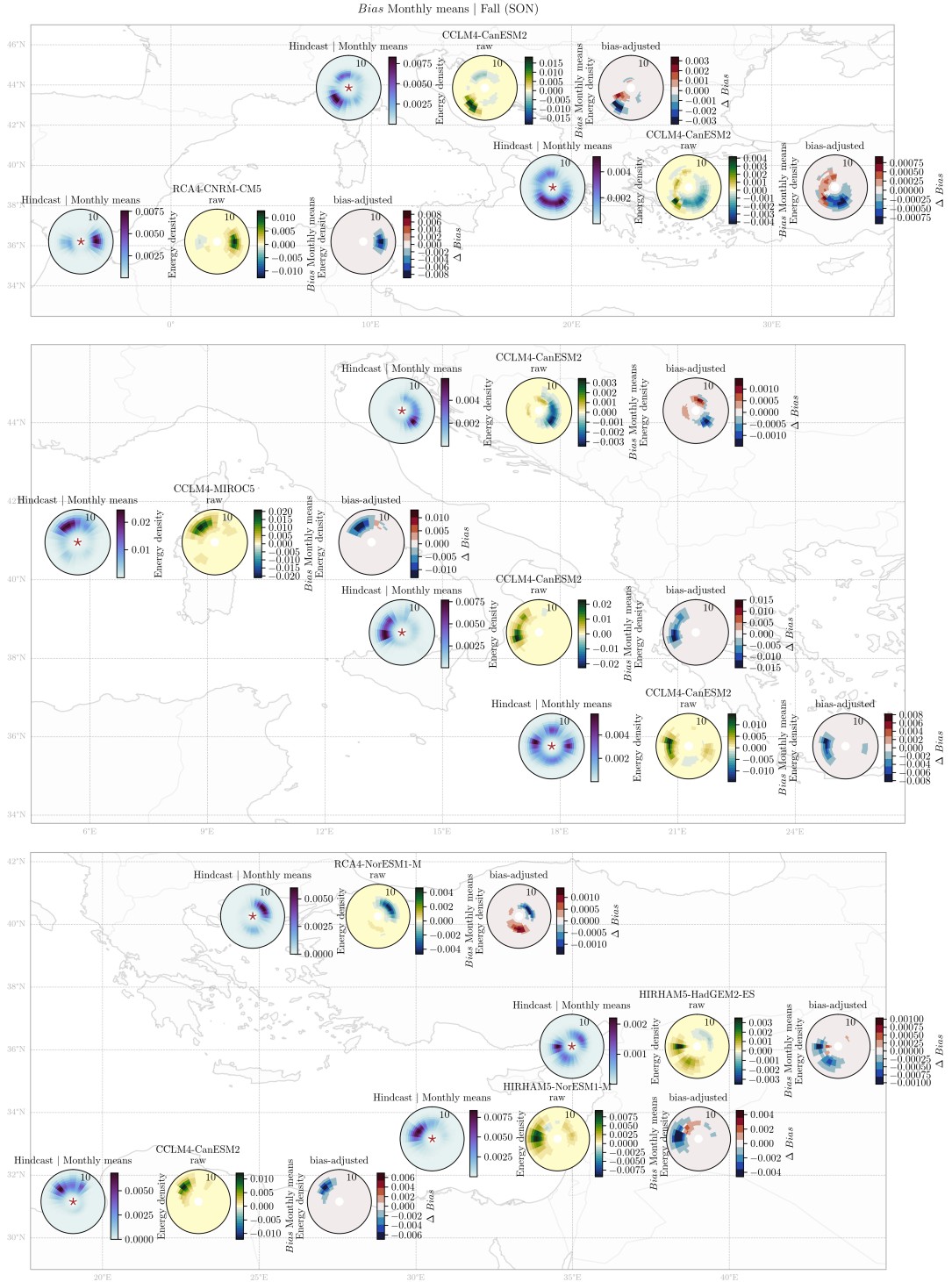

**Figure 7.** Monthly mean 2D spectra averaged over the fall (SON), in m$^2$s/deg. For each location: hindcast wave spectrum (left panel), $Bias$ between the raw worst-performing GCM-RCM and hindcast (middle) and $\Delta Bias$ between the bias-adjusted $Bias^*$ and raw $Bias$.

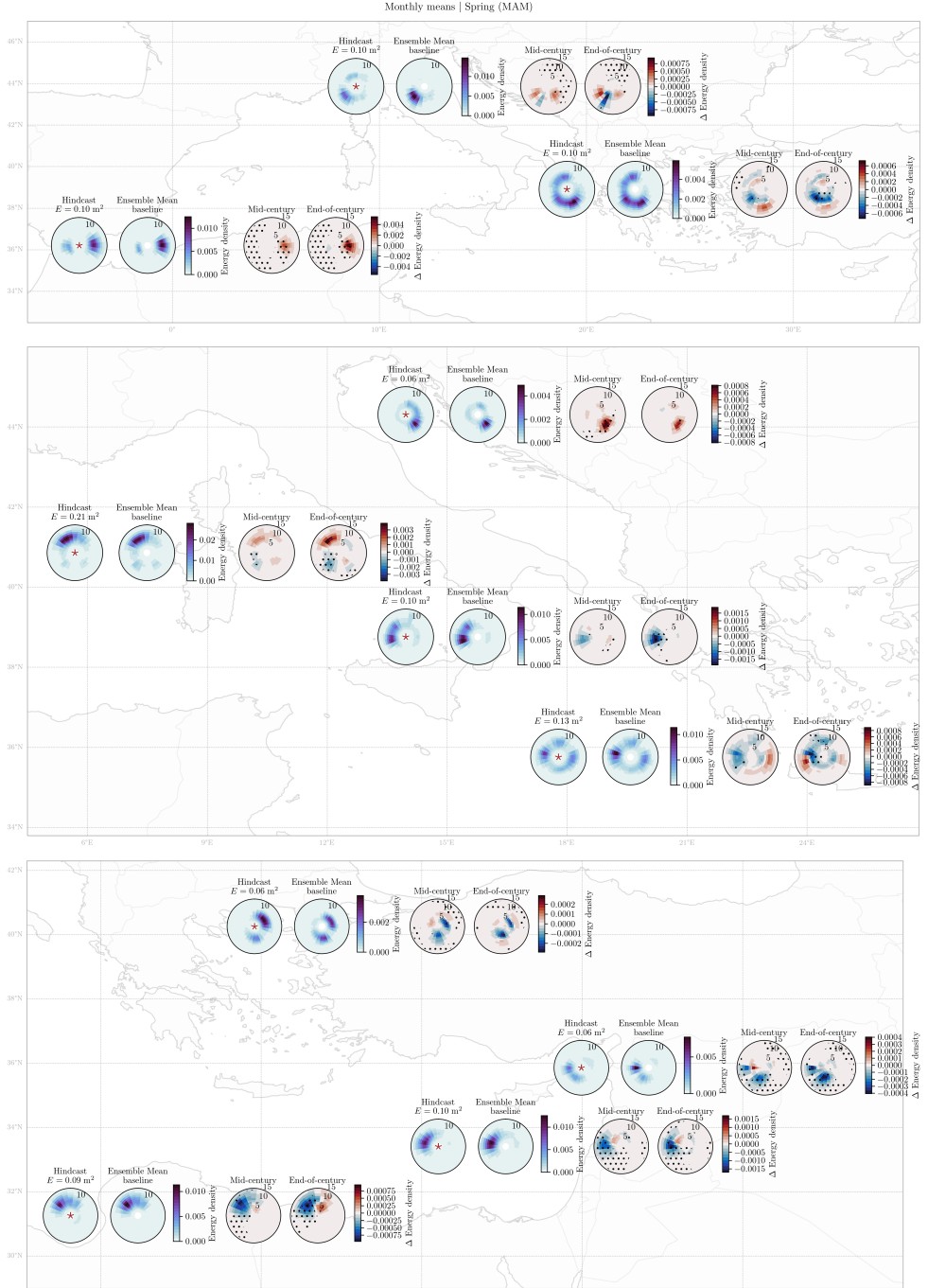

**Figure 8.** Spring (MAM) seasonal mean 2D spectra, in m$^2$s/deg. For each location: hindcast wave spectrum, ensemble mean for baseline conditions, changes between the multi-model bias-adjusted ensemble mean with respect to baseline period for mid-century $(2034 - 2060)$ and end-of-century conditions $(2074 - 2100)$, for all the analyzed locations in the Mediterranean Sea. Stippling indicates regions where at least 80% of the models agree on the sign of the change. Lack of stippling indicates low model agreement (less than 80%). The star in the hindcast/left panel represents the location of the points corresponding to Figure 1.

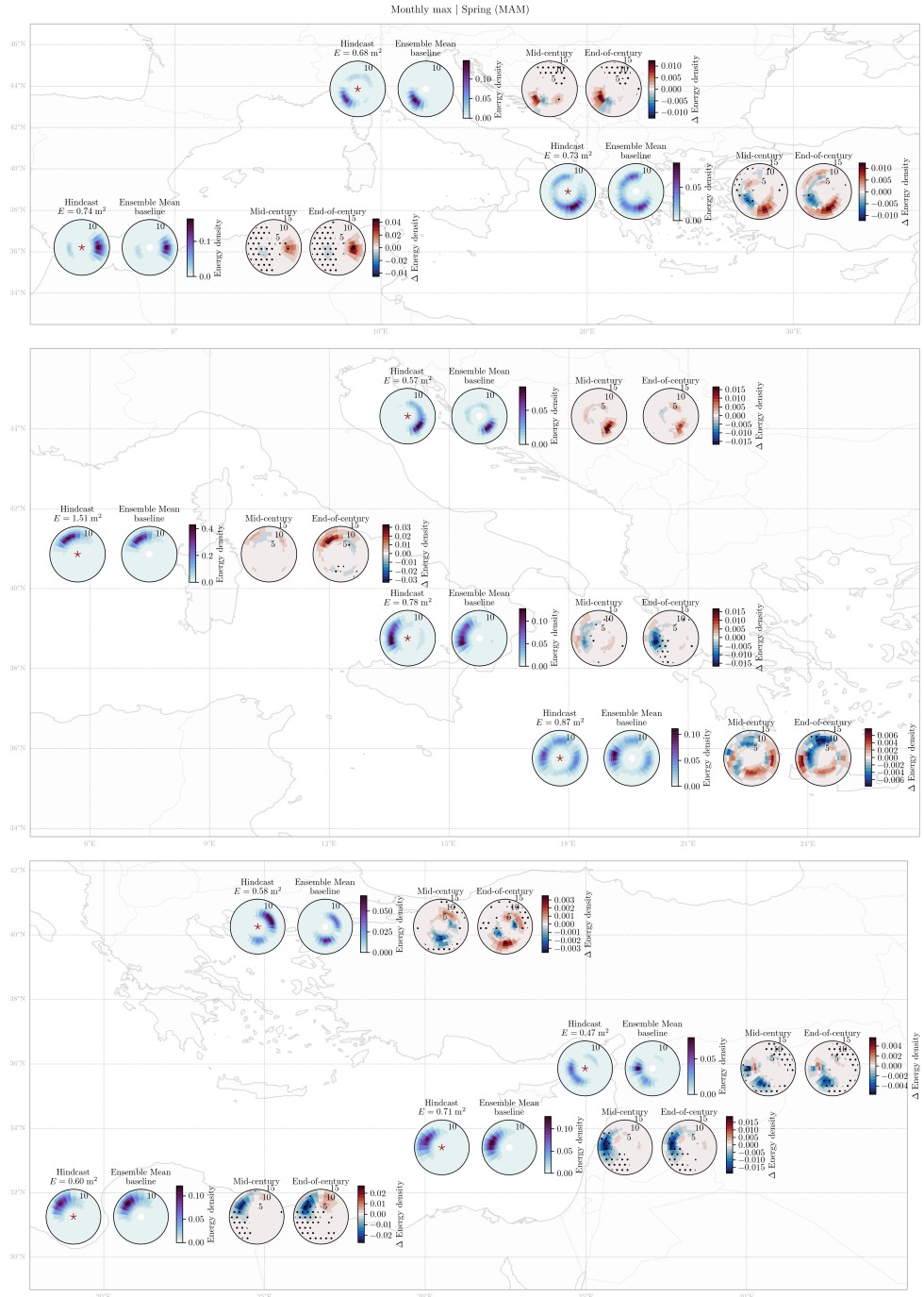

**Figure 9.** Monthly maxima 2D spectra averaged over the spring (MAM), in m$^2$s/deg. For each location: hindcast wave spectrum, ensemble mean for baseline, changes between the multi-model bias-adjusted ensemble mean with respect to baseline period for mid-century $(2034 - 2060)$ and end-of-century conditions $(2074 - 2100)$, for all the analyzed locations in the Mediterranean Sea. Stippling indicates regions where at least 80% of the models agree on the sign of the change. Lack of stippling indicates low model agreement (less than 80%).

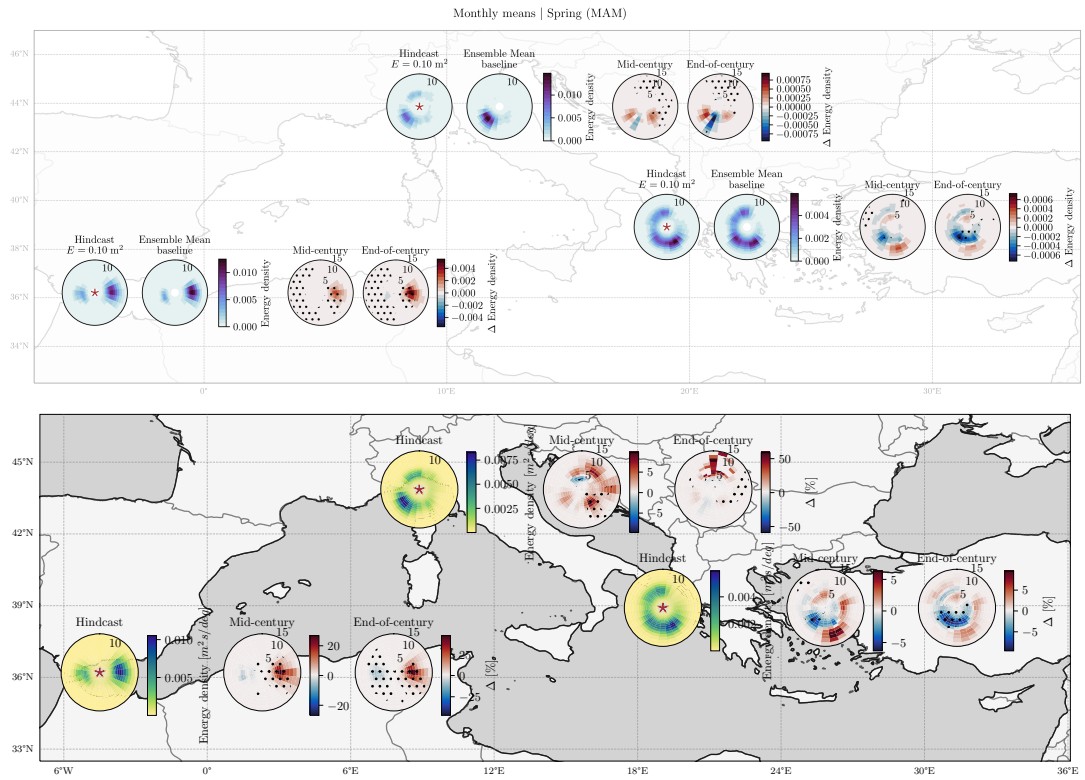

**Figure 10.** Mean directional spectra for spring (March, April and May). Top panel extracted from Figure 8, indicating the hindcast wave spectrum, ensemble mean for baseline conditions (1979–2005), changes between the multi-model bias-adjusted ensemble mean with respect to baseline period under RCP8.5 for mid-century (2034–2060) and end-of-century conditions (2074–2100). Bottom panel extracted from Figure 5 of Lira-Loarca and Besio (2022) representing, for each location, hindcast data and projected percent change for the multi-model ensemble for mid-century and end-of-century. In both panels, stippling indicates regions where at least 80% of models agree on the sign of change.

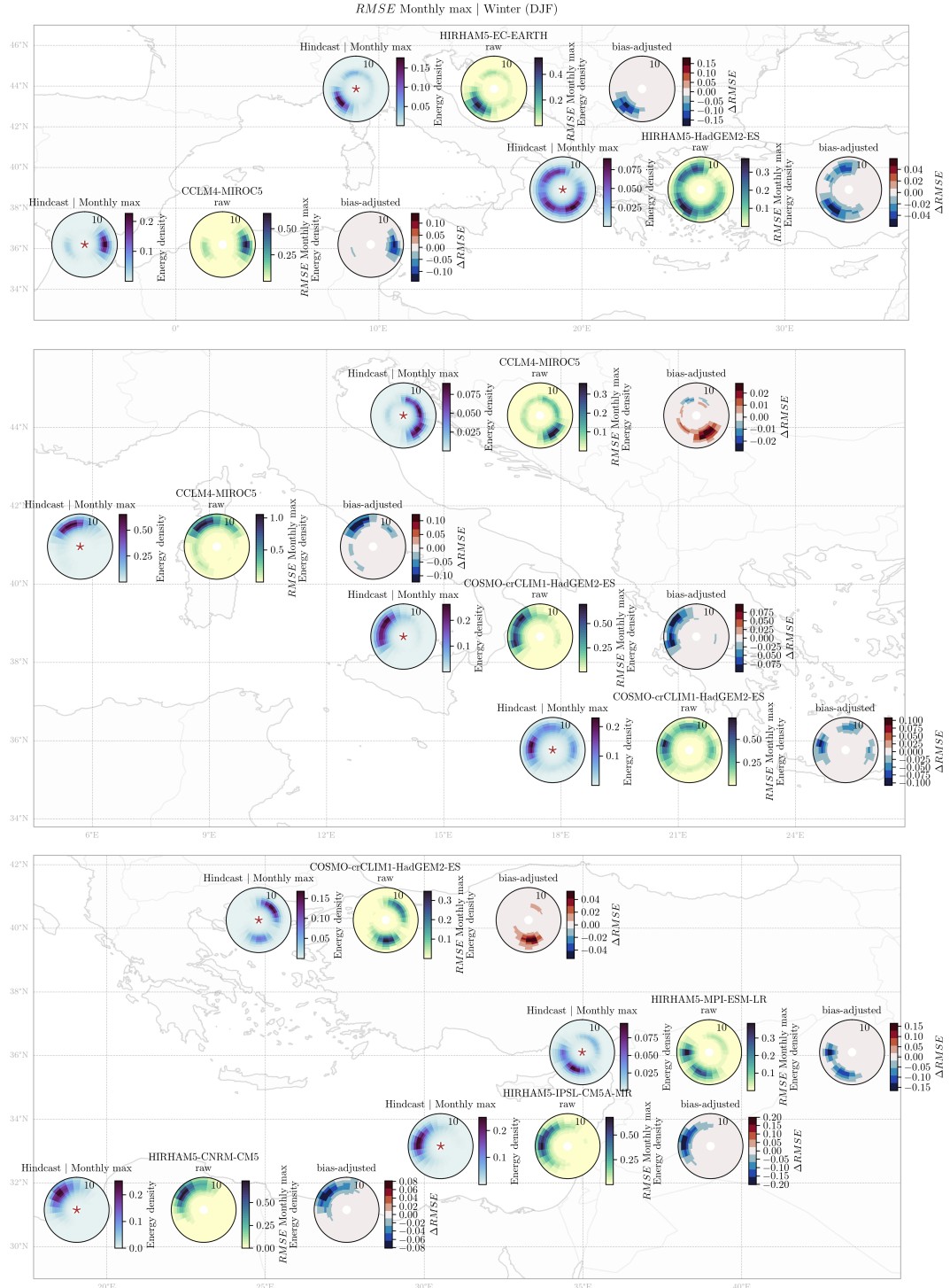

**Figure 11.** Results of the monthly maxima 2D spectra averaged over the winter (DJF). For each location: hindcast wave spectrum (left panel), $RMSE$ between the raw worst-performing GCM-RCM and hindcast (middle) and $\Delta RMSE$ between the bias-adjusted $RMSE^*$ and raw $RMSE$.