# Peer review of "Ocean wave spectra bias correction through energy conservation for climate change impacts"

_EGUsphere, 2024_

## Referee Comment (RC1)

General Comment:

This is a comprehensive study that deals with wave spectra bias correction methods and its applications for the future wave climate in a Mediterranean Sea. The study is well written, the novel methods looks appropriate and results are of high scientific value (although all figures need a revision). What I particularly liked in this study that new methods were developed and results are promising. However, I missed a broader discussion and comparison with the previous studies and differences in the results. I believe that this could add more value for this paper.

Also, for me was not clear why only the RCP8.5 was chosen for the future climate impacts. This is the most pessimistic scenario, therefore I would suggest to include at least one more scenario or prove, why only this was chosen.

The authors wanted to be short and presented only part of their results, however I think that all seasons bias correction results could be described in a paper and not moved to supplements.

Specific comments:

The weakest part of this paper is figures. Due to their complexity it is very hard to read. In a comment reply, I've found already updated figures. However, they all still need revision. At some figures text is hidden by circles or it is not readable due to the complex shoreline that disturbs a lot. Also, the legends should be homogenized, otherwise it sometimes misleads.

Figures in supplements should be revised, too. The legends are terrible, a lot of text on a figure makes it unreadable.

Abstract. Line 8

The end-of-century period is wrong, please correct it. 2064 or 2074?

Lines 40-44

Is this a study of Lira-Loarca, Besio of 2022 or 2023?

Lines: 168, 183, 212, 236

Typing errors

---

## Author Response (AR1)

EGUsphere [preprint], https://doi.org/10.5194/egusphere-2024-2947, 2024

**On the role of wave climate temporal variability in bias correction of wave climate projections**

Andrea Lira Loarca and Giovanni Besio

**RESPONSE TO REVIEWER 1**

The authors would like to thank the Reviewer for the comments, which have certainly improved the quality and readability of the manuscript. We have carefully considered all their suggestions and comments and have made corrections in the revised version of the manuscript. Below, you can find a detailed explanation on how each remark made by the reviewers has been incorporated.

Throughout the text, the reviewer's original comments are written in italics and after each of them, our response is given together with an explanation on how their comments have been accounted for.

*General Comment:*

*This is a comprehensive study that deals with wave spectra bias correction methods and its applications for the future wave climate in a Mediterranean Sea. The study is well written, the novel methods looks appropriate, and results are of high scientific value (although all figures need a revision). What I particularly liked in this study that new methods were developed, and results are promising. However, I missed a broader discussion and comparison with the previous studies and differences in the results. I believe that this could add more value for this paper.*

We appreciate the comment given by the reviewer. Regarding the comparison with previous studies, we believe the reviewer refers to the study by the same authors:

*Lira-Loarca, A. and Besio, G., 2022. Future changes and seasonal variability of the directional wave spectra in the Mediterranean Sea for the 21st century. Environmental Research Letters, 17(10), p.104015.*

The Lira-Loarca and Besio (2022) paper, used a bin-by-bin (frequency and direction) and seasonal-mean bias correction which had two major drawbacks:

- It did not allow for the reconstruction of the full time-series of bias corrected spectra since the adjustment was done directly to seasonal statistics.
- The energy conservation of the spectra was not guaranteed when each bin was adjusted independently
- The bin-by-bin method "cancelled" future wave systems on the GCM-RCMs that weren't present in the hindcast historical spectra.

The main novelty of our current study is the bias-correction technique which allows to correct the entire time series allowing for energy conservation and the preserving the 2D frequency and directional characteristics of the GCMs spectra for future conditions.

Therefore, the comparison with the current study is not straightforward as for example, the previous study did not include results on maximum values. It did, however, present the results of future changes in seasonal means. Therefore, we have updated the discussion of the manuscript following the reviewer's suggestion and included a comparison of the seasonal means for some locations.

*Also, for me was not clear why only the RCP8.5 was chosen for the future climate impacts. This is the most pessimistic scenario, therefore I would suggest to include at least one more scenario or prove, why only this was chosen.*

We appreciate this comment and understand the reviewer's concern about the use of a single climate scenario. To justify our choice, we would first like to emphasize that the scope of the present work is to provide a new bias-correction method for 2D wave spectra and the benefits of its use for an accurate characterization of future changes in wave spectra for a large ensemble of GCM-RCMs.

Regarding the use of RCP8.5 we would also like to point out:

- The RCP8.5 scenario, commonly defined as the "worst-case scenario", provided, up until the definition of the AR6 SSP-RCP scenarios, emissions in close agreement with historical total cumulative $CO_2$ emissions (within 1%). It is considered, within the AR5 RCP scenarios, as the best match for mid-century climate conditions under the last decade and current policies, providing high plausible values of $CO_2$ emissions in 2100.

Therefore, the scenario RCP8.5 is the focus by many coastal impact studies, above other RCP scenarios, providing useful and appropriate projections for quantifying physical climate risk, especially over near- to midterm policy-relevant time horizons. (IPCC, 2018, 2019; Schwalm et al, 2020; Simonetti et al. 2023). Some studies combine RCP8.5 with a more "moderate scenario" (Lionello et al., 2008; Lobeto et al., 2021) and obtain that both scenarios show the same patterns of change although with lower magnitude (leading to statistically insignificant changes in many regions). They mostly do not show different behaviour.

- The number of GCM-RCMs that have been simulated for RCP8.5 under EURO-CORDEX is larger than with any other climate scenario, thus allowing a statistically more robust representation of the projected changes, in this case, using an ensemble of 17 GCM-RCMs.

- Finally, the generation of the database with a large amount of wave simulations comes at a very high computational cost and data storage, that, in practice, prevents extending the calculation to a much larger number of models and/or scenarios.

The use of a single scenario, RCP8.5, is acknowledged as a limitation in the revised version of the manuscript following the reviewer's concern and recommendations for future work have been included.

References:
- IPCC, 2018: Global Warming of 1.5°C. An IPCC Special Report on the impacts of global warming of 1.5°C above pre-industrial levels and related global greenhouse gas emission pathways, in the context of strengthening the global response to the threat of climate change, sustainable development, and efforts to eradicate poverty [Masson-Delmotte, V., P. Zhai, H.-O. Pörtner, D. Roberts, J. Skea, P.R. Shukla, A. Pirani, W. Moufouma-Okia, C. Péan, R. Pidcock, S. Connors, J.B.R. Matthews, Y. Chen, X. Zhou, M.I. Gomis, E. Lonnoy, T. Maycock, M. Tignor, and T. Waterfield (eds.)].
- IPCC, 2019: IPCC Special Report on the Ocean and Cryosphere in a Changing Climate [H.-O. Pörtner, D.C. Roberts, V.Masson-Delmotte, P. Zhai, M. Tignor, E. Poloczanska, K. Mintenbeck, A. Alegría, M. Nicolai, A. Okem, J. Petzold, B. Rama, N.M. Weyer (eds.)].
- Lionello, P., Cogo, S., Galati, M., and Sanna, A. (2008). The mediterranean surface wave climate inferred from future scenario simulations. Global and Planetary Change 63, 152–162. Mediterranean climate: trends, variability and change
- Lobeto, H., Menendez, M., and Losada, I. J. (2021). Future behavior of wind wave extremes due to climate change. Scientific Reports 11, 7869.
- Simonetti, I. and Cappietti, L. (2023). Mediterranean coastal wave-climate long-term trend in climate change scenarios and effects on the optimal sizing of OWC wave energy converters. Coastal Engineering 179, 104247.
- Schwalm C. R., S. Glendon, P. B. Duffy. (2020) RCP8.5 tracks cumulative CO2 emissions. Proceedings of the National Academy of Sciences, 117 (33) 19656-19657; DOI: 10.1073/pnas.2007117117.

*The authors wanted to be short and presented only part of their results, however I think that all seasons bias correction results could be described in a paper and not moved to supplements.* Following the reviewer's suggestion, we have included and discussed the Bias' results for all seasons and the RMSE for winter. To maintain a concise manuscript, the RMSE results for spring, summer, and fall are provided in the Supplementary Information.

*Specific comments:*

*The weakest part of this paper is figures. Due to their complexity it is very hard to read. In a comment reply, I've found already updated figures. However, they all still need revision. At some figures text is hidden by circles or it is not readable due to the complex shoreline that disturbs a lot. Also, the legends should be homogenized, otherwise it sometimes misleads. Figures in supplements should be revised, too. The legends are terrible, a lot of text on a figure makes it unreadable.*

We have modified the format of all the figures and removed unnecessary text in order to improve the readability. The coastline has been kept in the background since we believe that the location of the analyzed points gives a lot of information regarding the spectral characteristics. We hope the revised figures are suitable for publication.

*Abstract. Line 8. The end-of-century period is wrong, please correct it. 2064 or 2074?*

It is 2074. We have corrected this.

*Lines 40-44. Is this a study of Lira-Loarca, Besio of 2022 or 2023?*

It refers to a work of 2022:

Lira-Loarca, A. and Besio, G., 2022. Future changes and seasonal variability of the directional wave spectra in the Mediterranean Sea for the 21st century. Environmental Research Letters, 17(10), p.104015.

*Lines: 168, 183, 212, 236. Typing errors*

The typos have been corrected.

EGUsphere [preprint], https://doi.org/10.5194/egusphere-2024-2947, 2024

**On the role of wave climate temporal variability in bias correction of wave climate projections**

Andrea Lira Loarca and Giovanni Besio

**RESPONSE TO REVIEWER 2**

The authors would like to thank the Reviewer for the comments, which have certainly improved the quality and readability of the manuscript. We have carefully considered all their suggestions and comments and have made corrections in the revised version of the manuscript. Below, you can find a detailed explanation on how each remark made by the reviewers has been incorporated.

Throughout the text, the reviewer's original comments are written in italics and after each of them, our response is given together with an explanation on how their comments have been accounted for.

*The paper presents the SEGDM-month method, a new bias-adjustment technique for 2D directional wave spectra. It effectively accounts for intra-annual variability and extreme wave distributions, which is a crucial advancement over traditional univariate corrections.*

*The study applies the method to 17 EURO-CORDEX regional climate models across 11 locations in the Mediterranean. This wide spatial coverage enhances the robustness of the findings. Correctly identifying and adjusting extreme values (upper-tail correction) is a key improvement. The method ensures energy conservation across frequency and direction, which is an essential requirement for accurate climate impact studies. The study follows a well-structured methodology, clearly explaining the datasets, correction process, and validation. The figures also effectively illustrate the improvements brought by the SEGDM-month method.*

*However, the paper mainly compares GCM-RCMs against hindcast data, but direct validation with observed wave spectra (e.g., buoy data, satellite data) is limited. I believe that including a direct comparison with observational datasets would strengthen the reliability of the bias-correction method.*

We agree with the reviewer that a direct validation with observations is crucial and would strengthen the results. However, regarding satellite data, although some products already exist, we could not find any that was suitable for our study mainly due to the temporal availability of data. More specifically, to our knowledge, the three main EU Copernicus Marine Service Products:

- WAVE_GLO_PHY_SPC_FWK_L3_NRT_014_002,
- WAVE_GLO_PHY_SPC_L3_MY_014_006, and,
- WAVE_GLO_PHY_SPC_L4_NRT_014_004,

provide data ranging since 2016, 2018 and 2021 and on a trajectory-basis therefore not allowing for a complete comparison of the hourly spectral dataset of our locations for the historical baseline period from 1970 until 2005.

Regarding the comparison with buoys, we searched the Puertos del Estado Spanish dataset, the Italian RON dataset and the Copernicus Marine Service and could not find buoy spectral information that matched our location, since most buoys are located in near-shore regions.

Nonetheless, this study uses, as baseline, a hindcast which has been calibrated and validated, for its standard statistics, against buoy and satellite data both in the regular 10km grid and the unstructured grid (Mentaschi et al 2013, 2015; Besio et al., 2016; Lira-Loarca et al., 2021).

References:
Mentaschi L, Besio G, Cassola F and Mazzino A, 2013 Developing and validating a forecast/hindcast system for the Mediterranean Sea J. Coast. Res. 65 1551–6
Mentaschi L, Besio G, Cassola F and Mazzino, A 2015 Performance evaluation of Wavewatch III in the Mediterranean Sea Ocean Modell. 90 82–94
Besio G, Mentaschi L and Mazzino A 2016 Wave energy resource assessment in the Mediterranean sea on the basis of a 35-year hindcast Energy 94 50-63
Lira-Loarca, A., Caceres-Euse, A., De-Leo, F. and Besio, G., 2022. Wave modeling with unstructured mesh for hindcast, forecast and wave hazard applications in the Mediterranean Sea. Applied Ocean Research, 122, p.103118.

*Further, while the statistical correction method is well-detailed, the study lacks a deep discussion of why biases occur in different sub-basins (e.g., Adriatic, Aegean, Western Mediterranean). Are biases primarily due to errors in regional wind fields? Or are they related to model resolution limitations? A more physical discussion would add value.*

This study addresses biases between hindcast and historical GCM-RCMs simulations. The wave simulations are carried out, for both cases, using the same wave generation and propagation

model Wavewatch III and the same grid and forced with similar resolution wind fields since the hindcast is forced with a 10km wind field grid and the GCM-RCMs forcing is a 0.11° grid (approx. 12.5km). Therefore, the biases present in the GCM-RCM against the hindcast, are all due to the accurate representation of wind fields and the parametrizations both the GCMs and the RCMs simulations and downscaling. Both Global and Regional Climate models (GCM, RCM) have systematic errors (biases) in their output. A single GCM or RCM simulation represents just one possible pathway of the climate system, influenced by uncertainties arising from unresolved processes and parameterizations, among other factors, and, also, the lack of data assimilation within GCM or RCMs simulations.

The fact that biases are different within the different sub-basins could be due to the mentioned systematic errors but also due to the accurate representation of small-scale phenomena, which is crucial in some sub-basins such as the Adriatic, Aegean and Ionian Sea. The different possible systematic errors have been addressed in the revised version of the manuscript.

*Then, the study highlights improvements but does not discuss potential pitfalls of the method. For example, does the bias correction impact the representation of low-energy wave conditions? Could the imposed energy conservation lead to unintended distortions in wave propagation physics?*

The SEGDM-month does not alter the representation of any wave energy system within the 2D spectra, since it does not re-distribute energy between bins but maintains the energy frequency and directional distribution given by the GCM-RCM signal and modifies only the integrated energy of the spectra. Therefore, it should not impact the representation of low or high energy conditions, since those are kept following the model signal.

Regarding the unintended distortions in wave propagation physics, it is important to emphasize that all the selected locations are considered in deep waters, with depths above the 800 m expect for the Adriatic that has 60 m, which is on the threshold between deep and intermediate waters for the typical wave periods occurring in the Adriatic basin, therefore, wave propagation physics are not into place. Indeed, the reviewer points out an interesting topic regarding the bias corrections of near-shore spectra. In those cases, yes, bias-correction could lead to unintended distortion in wave propagation physics since it could lead to an increase in energy that might not be physically feasible. Therefore, we have included this in the discussion, pointing out the need for research in near-shore regions and the use of the proposed method for intermediate depth only and its benefit to use the results as forcing for local wave and impact studies.

*Additionally, the study presents SEGDM-month as an improvement over previous methods (e.g., delta method), but a direct comparison with other advanced bias-correction techniques (e.g., quantile mapping used in atmospheric and hydrological modeling) is missing. I think it should also be discussed how the SEGDM-month methodology compares with emerging multivariate approaches to further quantify its advantages and limitations.*

The SEGDM-month method involves the use of the Empirical Gumbel Distribution Mapping (EGDM) method for the wave energy distribution on a monthly basis. The EGDM is a modified version of the "Quantile Mapping", where the upper-tail of the distribution is adjusted by a standard Gumbel distribution and the rest of the distribution, by Distribution Mapping (DM) using an Empirical Distribution therefore equivalent to the Empirical Quantile Mapping, Probability Mapping, or Quantile-Quantile mapping.

Therefore, the main differences between the SEGDM-month and a quantile mapping method would be the use of the Gumbel distribution for the upper-tail and the month-by-month adjustment. Regarding the former, Lemos *et al.*, 2020 proved that the EGQM provided better performance in correcting wave climate against the delta and EQM methods. Regarding the month-by-month distribution, Lira-Loarca *et al.*, 2023 highlighted the need of taking into account the temporal variability of waves and proved that using a simple EQM approach could lead to increased biases during summer conditions. These motivations have been highlighted in the revised version of the manuscript.

Although different multivariate bias-correction methods have emerged in literature, such as the N-dimensional probability density function, the dynamical Optimal Transport Correction, and the Rank Resampling for Distributions and Dependencies, among others, they all rely, on the first step, on adjusting different climate variables to their marginal univariate distribution (Allard *et al.*, 2024; Dieng *et al.*, 2022). In this study, we cannot really apply these methods given that the "multivariate" factor here is given by the 2D distribution of wave energy, in the directional and frequency space. To the authors' knowledge there is not any multivariate method that can be translated to correct 2D wave spectra.

References:
Allard, D., Vrac, M., François, B. and García de Cortázar-Atauri, I., 2024. Assessing multivariate bias corrections of climate simulations on various impact models under climate change. Hydrology and Earth System Sciences Discussions, 2024, pp.1-39.
Dieng, D., Cannon, A.J., Laux, P., Hald, C., Adeyeri, O., Rahimi, J., Srivastava, A.K., Mbaye, M.L. and Kunstmann, H., 2022. Multivariate bias-correction of high-resolution regional climate change simulations for West Africa: performance and climate change implications. Journal of Geophysical Research: Atmospheres, 127(5), p.e2021JD034836.
Lemos, G., Menendez, M., Semedo, A., Camus, P., Hemer, M., Dobrynin, M. and Miranda, P.M., 2020. On the need of bias correction methods for wave climate projections. Global and Planetary Change, 186, p.103109.
Lira Loarca, A., Berg, P., Baquerizo, A. and Besio, G., 2023. On the role of wave climate temporal variability in bias correction of GCM-RCM wave simulations. Climate Dynamics, 61(7), pp.3541-3568.

*Finally, the paper assesses bias correction but does not provide much discussion on how corrected wave projections alter climate impact assessments. How does the correction affect key climate adaptation strategies, such as coastal erosion predictions or marine energy assessments?*
We have included in the discussion of the revised version of the manuscript comparing the results of future changes in seasonal mean wave spectra between the SEGDM-month in this work and the delta-method used by the same authors in the work Lira-Loarca et al., 2022, to depict how different bias-correction techniques could lead to different assessments of projected changes and therefore, different climate adaptation strategies.

References:
Lira-Loarca, A. and Besio, G., 2022. Future changes and seasonal variability of the directional wave spectra in the Mediterranean Sea for the 21st century. Environmental Research Letters, 17(10), p.104015.

*In brief, it seems that the SEGDM-month method offers a substantial advancement in bias correction for 2D directional wave spectra by integrating energy conservation, extreme value adjustment via the Gumbel distribution, and monthly corrections to account for seasonal variability. Compared to traditional univariate and simpler delta methods, it better preserves the complex interdependencies of the wave spectrum. However, in my opinion, challenges remain—particularly regarding validation in low-energy conditions, reliance on high-quality hindcast data, and performance in regions with strong local-scale phenomena. Consequently, I*

*would really like the authors to further discussed their results as I believe it would strengthen the study.*

The authors appreciate the comments by the reviewer and hope we have answered all the concerns in a satisfactory manner in the responses above, and that the revised version of the manuscript presents the requested improvements.